# Variance-Reduced Stochastic Gradient Descent on Streaming Data

**Ellango Jothimurugesan**[*][†]
Carnegie Mellon University
ejothimu@cs.cmu.edu

**Ashraf Tahmasbi**[*][‡]
Iowa State University
tahmasbi@iastate.edu

**Phillip B. Gibbons**[†]
Carnegie Mellon University
gibbons@cs.cmu.edu

**Srikanta Tirthapura**[‡]
Iowa State University
snt@iastate.edu

## Abstract

We present an algorithm STRSAGA that can efficiently maintain a machine learning model over data points that arrive over time, and quickly update the model as new training data are observed. We present a competitive analysis that compares the sub-optimality of the model maintained by STRSAGA with that of an offline algorithm that is given the entire data beforehand. Our theoretical and experimental results show that the risk of STRSAGA is comparable to that of an offline algorithm on a variety of input arrival patterns, and its experimental performance is significantly better than prior algorithms suited for streaming data, such as SGD and SSVRG.

## 1 Introduction

We consider the maintenance of a model over streaming data that are arriving as an endless sequence of data points. At any point in time, the goal is to fit the model to the training data points observed so far, in order to accurately predict/label unobserved test data. Such a model is never "complete" but instead needs to be continuously updated as newer training data points arrive. Methods that recompute the model from scratch upon the arrival of new data points are infeasible due to their high computational costs, and hence we need methods that *efficiently update* the model as more data arrive. Such efficiency should not come at the expense of accuracy—the accuracy of the model maintained through such updates should be close to that obtained if we were to build a model from scratch, using all the training data points seen so far.

Fitting a model is usually cast as an optimization problem, where the model parameters are those that optimize an objective function. In typical cases, the objective function is the empirical or regularized risk, usually the sum of a finite number of terms, and often assumed to be convex. Consider a stream of training data points $\mathcal{S}_i$ arriving before or at time $i$ consisting of $n_i$ data points. Let $\mathbf{w}$ denote the set of parameters characterizing the learned function. The empirical risk function $\mathcal{R}_{\mathcal{S}_i}$ measures the average loss of $\mathbf{w}$ over $\mathcal{S}_i$: $\mathcal{R}_{\mathcal{S}_i}(\mathbf{w}) = \frac{1}{n_i} \sum_{j=1}^{n_i} f_j(\mathbf{w})$, where $f_j(\mathbf{w})$ is the loss of $\mathbf{w}$ on data point $j$. The goal is to find the empirical risk minimizer (ERM), i.e., the parameters $\mathbf{w}^*$ that minimize the empirical risk over all data points observed so far. Typically, some form of gradient descent is used in pursuit of $\mathbf{w}^*$.

There are two common approaches: *batch learning* and *incremental learning* (sometimes called "online learning") [BL03, Ber16]. Batch learning uses all available data points in the training set to

---

[*]EJ and AT contributed equally to this work.

[†]Supported in part by NSF grant 1725663

[‡]Supported in part by NSF grants 1527541 and 1725702

compute the gradient for each step of gradient descent—this method renders a gradient computation to be expensive, especially for a large dataset. In contrast, an incremental learning algorithm operates on only a single data point at each step of gradient descent, and hence a single step of an incremental algorithm is much faster than a corresponding step of a batch algorithm. Incremental algorithms, e.g., Stochastic Gradient Descent (SGD) [RM51, BL03] and variance-reduced improvements such as SVRG [JZ13] and SAGA [DBLJ14], have been found to be more effective on large datasets than batch algorithms, and are widely used.

Both batch and incremental algorithms assume that all training data points are available in advance—we refer to such algorithms as *offline algorithms*. However, in the setting that we consider, data points arrive over time according to an unknown arrival distribution, and neither batch nor incremental algorithms are able to update the model efficiently as more data arrives. Though incremental learning algorithms use only a single data point in each iteration, they typically select that point from the entire set of training data points—this set of training data points is constantly changing in our setting, rendering incremental algorithms inapplicable. In the rest of the paper, we refer to an algorithm that can efficiently update a model upon the arrival of new training data points as a *streaming data algorithm*. Note that streaming data algorithms (which are not limited in their memory usage) are broader than traditional streaming algorithms (which work in a single pass with limited memory). Streaming data algorithms are relevant in many practical settings given the abundance of memory these days.

The optimization goal of a streaming data algorithm is to maintain a model using all the data points that have arrived so far, such that the model's empirical risk is close to the ERM over those data points. The challenges include (i) because the training data is changing at each time step, the ERM on streaming data is a "moving target"; (ii) the ERM is an optimal solution that cannot be realized in limited processing time, while a streaming data algorithm is not only limited in processing time, but is also presented the data points only sequentially; (iii) with increasing arrival rates, it becomes increasingly difficult for the streaming data algorithm to keep up with the ERM; and (iv) data points may not arrive at a steady rate: the numbers of points arriving at different points in time can be highly skewed. We present and analyze a streaming data algorithm, STRSAGA, that overcomes these challenges and achieves an empirical risk close to the ERM in a variety of settings.

**Contributions.** We present STRSAGA, a streaming data algorithm for maintaining a model. STRSAGA sees data points in a sequential manner, and can efficiently incorporate newly arrived data points into the model. Yet, its accuracy at each point in time is comparable to that of an offline algorithm that has access to all training data points in advance. We prove this using a "competitive analysis" framework that compares the accuracy of STRSAGA to a state-of-the-art offline algorithm DYNASAGA [DLH16], which is based on variance-reduced SGD. We show that given the same computational power, the accuracy of STRSAGA is competitive to DYNASAGA at each point in time, under certain conditions on the schedule of input arrivals. Our notion of "risk-competitiveness" is based on the sub-optimality of risk with respect to the ERM ("sub-optimality" is defined in Section 3). Our theoretical analysis relies on a connection between the "effective sample size" of the algorithm (whether a streaming data algorithm or an offline algorithm) and its sub-optimality of risk with respect to the ERM. We show that if a streaming data algorithm is "sample-competitive" to an offline algorithm, i.e., its effective sample size is close to that of an offline algorithm, then it is also risk-competitive to the offline algorithm.

A key aspect of our work is that we carefully consider the *schedule* of arrivals of the data points—we care not only about which training data points have arrived so far, and how many of them, but also about when they arrived. In our setting where the streaming data algorithm is computationally bounded, it is not possible to be always risk-competitive with an offline algorithm. However, we show that it is possible to achieve risk-competitiveness, if the schedule of arrivals of training data points obeys certain conditions that we lay out in Section 5. We show that these conditions are satisfied by a number of common arrival distributions, including Poisson arrivals and many classes of skewed arrivals. For all these arrival distributions, we show that STRSAGA is risk-competitive to DYNASAGA.

Our experimental results for two machine learning tasks, logistic regression and matrix factorization, on two real data sets each, support our analytical findings: the sub-optimality of STRSAGA on data points arriving over time (according to a variety of input arrival distributions) is almost always comparable to the offline algorithm DYNASAGA that is given all data points in advance, when each algorithm is given the same computational power. We also show that STRSAGA significantly outperforms natural

streaming data versions of both SGD and `SSVRG` [FGKS15]. Moreover, the update time of `STRSAGA` is small, making it practical even for settings when the arrival rate is high.

## 2 Related work

Stochastic Gradient Descent (SGD) [RM51] and its extensions are used extensively in practice for learning from large datasets. While an iteration of SGD is cheap relative to an iteration of a full gradient method, its variance can be high. To control the variance, the learning rate of SGD must decay over time, resulting in a sublinear convergence rate. Newer variance-reduced versions of SGD, on the other hand, achieve linear convergence on strongly convex objective functions, generally by incorporating a correction term in each update step that approximates a full gradient, while still ensuring each iteration is efficient like SGD.

SAG [RSB12] was among the first variance reduction methods proposed and achieves linear convergence rate for smooth and strongly convex problems. SAG requires storage of the last gradient computed for each data point and uses their average in each update. SAGA [DBLJ14] improves on SAG by eliminating a bias in the update. Stochastic Variance-Reduced Gradient (SVRG) [JZ13] is another variance reduction method that does not store the computed gradients, but periodically computes a full-data gradient, requiring more computation than SAGA. Semi-Stochastic Gradient Descent (S2GD) [KR13] is a variant of SVRG where the gaps between full-data gradient computations are of random length and follow a geometric law. CHEAPSVRG [SAKS16] is another variant of SVRG. In contrast with SVRG, it estimates the gradient through computing the gradient on a subset of training data points rather than all the data points. However, all of the above variance-reduced methods require $O(n \log n)$ iterations to guarantee convergence to statistical accuracy (to yield a good fit to the underlying data) for $n$ data points. `DYNASAGA` [DLH16] achieves statistical accuracy in only $O(n)$ iterations by using a gradually increasing sample set and running SAGA on it.

So far, all the algorithms we have discussed are offline, and assume the entire dataset is available beforehand. Streaming SVRG (`SSVRG`) [FGKS15] is an algorithm that handles streaming data arrivals, and processes them in a single pass through data, using limited memory. In our experimental study, we found `STRSAGA` to be significantly more accurate than `SSVRG`. Further, our analysis of `STRSAGA` shows that it handles arrival distributions which allow for burstiness in the stream, while `SSVRG` is not suited for this case. In many practical situations, restricting a streaming data algorithm to use limited memory is overly restrictive and as our results show, leads to worse accuracy.

## 3 Model and preliminaries

We consider a data stream setting in which the training data points arrive over time. For $i = 1, 2, 3, \dots$, let $\mathbf{X}_i$ be the set of zero or more training data points arriving at time step $i$. We assume that each training data point is drawn from a fixed distribution $\mathcal{P}$, which is not known to the algorithm. Dealing with distributions that change over time is beyond the scope of this paper. Let $\mathcal{S}_i = \cup_{j=1}^{i} \mathbf{X}_j$ denote the set of data points that have arrived in time steps 1 through $i$ (inclusive). Let $n_i$ denote the number of data points in $\mathcal{S}_i$.

The model being trained/maintained is drawn from a class of functions $\mathcal{F}$. A function in this class is parameterized by a vector of weights $\mathbf{w} \in \mathbb{R}^d$. For a function $\mathbf{w}$, we define its expected risk as $\mathcal{R}(\mathbf{w}) = \mathbb{E}\left[f_{\mathbf{x}}(\mathbf{w})\right]$ where $f_{\mathbf{x}}(\mathbf{w})$ is the loss of function $\mathbf{w}$ on input $\mathbf{x}$ and the expectation is taken over $\mathbf{x}$ drawn from distribution $\mathcal{P}$. Let function $\mathbf{w}^* = \arg\min_{\mathbf{w} \in \mathcal{F}} \mathcal{R}(\mathbf{w})$ denote the optimal function with respect to $\mathcal{R}(\mathbf{w})$. Let $\mathcal{R}^* = \mathcal{R}(\mathbf{w}^*)$ denote the minimum expected risk possible over distribution $\mathcal{P}$, within function class $\mathcal{F}$. The function $\mathbf{w}^*$ is called the *distributional risk minimizer*. Given a sample $\mathcal{S}$ of training data points drawn from $\mathcal{P}$, the best we can do is minimize the empirical risk over this sample. We have analogous definitions for minimizers of empirical risk over this sample. The empirical risk of function $\mathbf{w}$ over a sample $\mathcal{S}$ of $n$ elements is: $\mathcal{R}_{\mathcal{S}}(\mathbf{w}) = \frac{1}{n}\sum_{\mathbf{x} \in \mathcal{S}} f_{\mathbf{x}}(\mathbf{w})$. The optimizer of the empirical risk is denoted as $\mathbf{w}_{\mathcal{S}}^*$, defined as $\mathbf{w}_{\mathcal{S}}^* = \arg\min_{\mathbf{w} \in \mathcal{F}} \mathcal{R}_{\mathcal{S}}(\mathbf{w})$. The optimal empirical risk is $\mathcal{R}_{\mathcal{S}}^* = \mathcal{R}_{\mathcal{S}}(\mathbf{w}_{\mathcal{S}}^*)$. We denote the optimizer of the empirical risk over $\mathcal{S}_i$ as $\mathbf{w}_i^* = \mathbf{w}_{\mathcal{S}_i}^*$. Similarly, the optimal empirical risk over $\mathcal{S}_i$ is $\mathcal{R}_i^* = \mathcal{R}_{\mathcal{S}_i}(\mathbf{w}_i^*)$.

Suppose a risk minimization algorithm is given a set of training examples $\mathcal{S}_i$, and outputs approximate solution $\mathbf{w}_i$. The *statistical error* is $\mathbb{E}\left[\mathcal{R}(\mathbf{w}_i^*) - \mathcal{R}(\mathbf{w}^*)\right]$ and the *optimization error* is

$\mathbb{E}\left[\mathcal{R}(\mathbf{w}_i) - \mathcal{R}(\mathbf{w}_i^*)\right]$, where the expectation is taken over the randomness of $\mathcal{S}_i$. The *total error* (restricting to a fixed function class $\mathcal{F}$) is the sum of the two.

Following Bottou and Bousquet [BB07], we define *sub-optimality* as follows.

**Definition 1.** *The sub-optimality of an algorithm $A$ over training data $\mathcal{S}$ is the difference between $A$'s empirical risk and the optimal empirical risk:*

$$\texttt{SUBOPT}_{\mathcal{S}}(A) := \mathcal{R}_{\mathcal{S}}(\mathbf{w}) - \mathcal{R}_{\mathcal{S}}(\mathbf{w}_{\mathcal{S}}^*)$$

*where $\mathbf{w}$ is the solution returned by $A$ on $\mathcal{S}$ and $\mathbf{w}_{\mathcal{S}}^*$ is the empirical risk minimizer over $\mathcal{S}$.*

Let $\mathcal{H}(n) = cn^{-\alpha}$, for a constant $c$ and $1/2 \leq \alpha \leq 1$, be an upper bound on the statistical error. Bottou and Bousquet [BB07] show that if $\epsilon$ is a bound on the sub-optimality of $\mathbf{w}$ on $\mathcal{S}_i$, then the total error is bounded by $\mathcal{H}(n_i) + \epsilon$. Therefore, in designing an efficient algorithm for streaming data, we focus on reducing the sub-optimality to asymptotically balance with $\mathcal{H}(n_i)$ — it does not pay to reduce the empirical risk even further. Note that although $\mathcal{H}(n_i)$ is only an upper bound on the statistical error, Bottou and Bousquet remark "it is often accepted that these upper bounds give a realistic idea of the actual convergence rates" [BB07], in which case balancing the sub-optimality with $\mathcal{H}(n_i)$ asymptotically minimizes the total error.

We focus on time-efficient algorithms for maintaining a model over streaming data. We focus on a basic step used in all SGD-style algorithms (or variants such as SAGA): A random training point $x$ is chosen from a set of training samples, and the vector $\mathbf{w}$ is updated through a gradient computed at point $x$. Let $\rho \geq 1$ denote the number of such basic steps that can be performed in a single time step.

# 4   STRSAGA: gradient descent over streaming data

We present our algorithm STRSAGA for learning from streaming data. Stochastic gradient descent (or one of its variants, such as SAGA [DBLJ14]) works by repeatedly sampling a point from a training set $T$ and using its gradient to determine an update direction. One option to handle streaming data arrivals is to simply expand the set $T$ from which further sampling is conducted, by adding all the new arrivals. However, the problem with this approach is that the size of the training set $T$ can change in an uncontrolled manner, depending on the number of arrivals. As illustrated in prior work [DLH16], the optimization error of SAGA increases with the size of the training set $T$. With an uncontrolled increase in the size of $T$, the corresponding sub-optimality of the algorithm over $T$ increases, so that the function that is finally computed may have poor accuracy.

To handle this, we use an idea from DYNASAGA [DLH16], which increases the size of the training set $T$ in a controlled manner, according to a schedule. Upon increasing the size of $T$, further increases are placed on hold until a sufficient number of SAGA steps have been performed on the current state of $T$. By using this idea, DYNASAGA was able to achieve statistical accuracy earlier than SAGA. However, DYNASAGA is still an offline algorithm that assumes that all training data is available in advance.

STRSAGA deals with streaming arrivals as follows. Arriving points from the next set of points $\mathbf{X}_i$ are added to a buffer Buf. The effective sample set $T$ is expanded in a controlled manner, similar to DYNASAGA. However, instead of choosing new points from a static training set, such as in DYNASAGA, STRSAGA chooses new points from the dynamically changing buffer Buf. If Buf is empty, then available CPU cycles are used to perform further steps of SAGA. After any time step, it is possible that STRSAGA may have trained over only a subset of the points that are available in Buf, but this is to ensure that the optimization error on the subset that has been trained is balanced with the statistical error of the effective sample size. Algorithm 1 depicts the steps taken to process the zero or more points $\mathbf{X}_i$ arriving at time step $i$. Before any input is seen, the algorithm initializes buffer Buf to empty, effective sample $T_0$ to empty, and function $\mathbf{w}_0$ to random values. STRSAGA as described here uses the basic framework of DYNASAGA, of adding one training point to $T_i$ every two steps of SAGA (the linear schedule in [DLH16]), and both algorithms borrow variance-reduction steps from SAGA (lines 8-9 in Algorithm 1 and using the running average $A$ of all gradients).

**Analysis of STRSAGA:** Suppose data points $\mathcal{S}_i$ have been seen till time step $i$, and $n_i = |\mathcal{S}_i|$. We first note that the time taken to process a set of training points $\mathbf{X}_i$ is dominated by the time taken for $\rho$ iterations of SAGA. Ideally, the empirical risk of the solution returned by STRSAGA is close to the empirical risk of the ERM over $\mathcal{S}_i$. However, this is not possible in general. Suppose the number

---

**Algorithm 1:** STRSAGA: Process a set of training points $\mathbf{X}_i$ that arrived in time step $i$, $i > 0$.

---
```
// w_{i-1} is the current function.  T_{i-1} the effective sample set.
```
1 Add $\mathbf{X}_i$ to Buf   `// Buf is the set of training points not added to` $T_i$ `yet`
2 $\widetilde{\mathbf{w}}_0 \leftarrow \mathbf{w}_{i-1}$ and $T_i \leftarrow T_{i-1}$
```
   // Do ρ steps of SAGA at each time step
```
3 **for** $j \leftarrow 1$ **to** $\rho$ **do**
```
      // Every two steps of SAGA, add one training point to T_i, if available
```
4    **if** (Buf *is non-empty) AND (j is even)* **then**
5      Move a single point, $z$, from Buf to $T_i$
6      $\alpha(z) \leftarrow 0$   `//` $\alpha(z)$ `the prior gradient of` $z$`, initialized to 0`
7      $A \leftarrow \sum_{p \in T_i} \alpha(p)/|T_i|$   `//` $A$ `the average of all gradients, used by`
              `SAGA, and can be maintained incrementally`
8    Sample a point $p$ uniformly from $T_i$
9    $g \leftarrow \nabla f_p(\widetilde{\mathbf{w}}_{j-1})$   `// compute the gradient`
10    $\widetilde{\mathbf{w}}_j \leftarrow \widetilde{\mathbf{w}}_{j-1} - \eta(g - \alpha(p) + A)$   `//` $\eta$ `is the learning rate`
11    $\alpha(p) \leftarrow g$
12 $\mathbf{w}_i \leftarrow \widetilde{\mathbf{w}}_\rho$
```
   // w_i is the current function and T_i is the effective sample set.
```

---

of points arriving at each time step $i$ were much greater than $\rho$, the number of iterations of SAGA that can be performed at each step. Then not even an offline algorithm such as DYNASAGA that has all points at the beginning of time could be expected to match the empirical risk of the ERM within the available time. In what follows, we present a competitive analysis, where the performance of STRSAGA is compared with that of an offline algorithm that has all data available to it in advance. We consider two offline algorithms, ERM and DYNASAGA($\rho$), described below.

**Algorithm ERM** is the empirical risk minimizer, sees all of $\mathcal{S}_i$ at the beginning of time, and has infinite computational power to process it. A streaming data algorithm has two obstacles if it has to compete with ERM: (i) Unlike ERM, a streaming data algorithm does not have all data in advance, and (ii) Unlike ERM, a streaming data algorithm has limited computational power. It is clear that no streaming data algorithm can do better than ERM. We can practically approach the performance of ERM through executing DYNASAGA until convergence is achieved.

**Algorithm DYNASAGA($\rho$)** sees all of $\mathcal{S}_i$ at the beginning of time, and is given $\rho$ iterations of gradient computations in each step. The parenthetical $\rho$ denotes this algorithm is the extension of the original DYNASAGA [DLH16], parameterized by the available amount of processing time. The algorithm DYNASAGA performs $2n_i$ steps of gradient computations on $\mathcal{S}_i$ and then terminates, while DYNASAGA($\rho$) performs $\rho i$ steps, where if $\rho i > 2n_i$, the additional steps are uniformly over $\mathcal{S}_i$. The computational power of DYNASAGA($\rho$) over $i$ time steps matches that of a streaming data algorithm. However, DYNASAGA($\rho$) is still more powerful than a streaming data algorithm, because it can see all data in advance. In general, it is not possible for a streaming data algorithm to compete with DYNASAGA($\rho$) either—one issue being that streaming arrivals may be very bursty. Consider the extreme case when all of $\mathcal{S}_i$ arrives in the $i$th time step, and there were no arrivals in time steps 1 through $i-1$. An algorithm for streaming data has only $\rho$ gradient computation steps that it can perform on $n_i$ points, and its earlier $\rho(i-1)$ gradient steps had no data to use. In contrast, DYNASAGA($\rho$) can perform $\rho i$ gradient steps on $\mathcal{S}_i$, and achieve a smaller empirical risk.

Each algorithm STRSAGA, DYNASAGA($\rho$), and ERM, after seeing $\mathcal{S}_i$, has trained its model on a subset $T_i \subseteq \mathcal{S}_i$. We call this subset the "effective sample set". Let $t_i^{\text{STR}}, t_i^D$, and $t_i^{\text{ERM}}$ denote the sizes of the effective sample sets of STRSAGA, DYNASAGA($\rho$), and ERM, respectively, after $i$ time steps. The following lemma shows that the expected sub-optimality of DYNASAGA($\rho$) over $\mathcal{S}_i$ is related to $t_i^D$.

**Lemma 1** (Lemma 5 in [DLH16]). *After $i$ time steps, $t_i^D = \min\{n_i, \rho i/2\}$, and $t_i^{ERM} = n_i$. The expected sub-optimality of* DYNASAGA*($\rho$) over $\mathcal{S}_i$ after $i$ time steps is $O(\mathcal{H}(t_i^D))$.*

Our goal is for a streaming data algorithm to achieve an empirical risk that is close to the risk of an offline algorithm. We present our notion of risk-competitiveness in Definition 2.

**Definition 2.** *For $c \geq 1$, a streaming data algorithm $I$ is said to be c-risk-competitive to* DYNASAGA*($\rho$) at time step $i$ if $\mathbb{E}\left[\text{SUBOPT}_{\mathcal{S}_i}(I)\right] \leq c\mathcal{H}(t_i^D)$. Similarly, $I$ is said to be c-risk-competitive to ERM at time step $i$ if $\mathbb{E}\left[\text{SUBOPT}_{\mathcal{S}_i}(I)\right] \leq c\mathcal{H}(n_i)$.*

Note that the expected sub-optimality of $I$ is compared with $\mathcal{H}(t_i^D)$ and $\mathcal{H}(n_i)$, which are upper bounds on the statistical errors of DYNASAGA($\rho$) and ERM respectively. If $\mathcal{H}()$ is a tight bound on the statistical error, and hence, a lower bound on the total error, then $c$-risk-competitiveness to DYNASAGA($\rho$) implies that the expected sub-optimality of the algorithm $I$ is within a factor of $c$ of the total risk of DYNASAGA($\rho$), as illustrated in Figure 1. We next show if a streaming data algorithm is risk-competitive with respect to DYNASAGA($\rho$) then it is also risk-competitive with respect to ERM, under certain conditions.

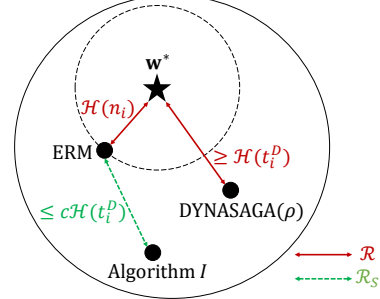

Figure 1: The error of each algorithm.

**Lemma 2.** *If a streaming data algorithm $I$ is c-risk-competitive to* DYNASAGA*($\rho$) at time step $i$, and the statistical risk $\mathcal{H}(n) = n^{-\alpha}$, then $I$ is $c \cdot \max\left(\left(\frac{2\widetilde{\lambda}_i}{\rho}\right)^{\alpha}, 1\right)$-risk-competitive to ERM at time step $i$, where $\widetilde{\lambda}_i = \left(\frac{n_i}{i}\right)$ and $n_i$ is the size of $\mathcal{S}_i$.*

*Proof.* From Definition 2 we have: $\mathbb{E}\left[\text{SUBOPT}_{\mathcal{S}_i}(I)\right] \leq c\mathcal{H}(t_i^D)$. We know $t_i^D = \min(n_i, \rho i/2)$ (Lemma 1). First consider the case when $n_i \leq \rho i/2$. We have: $\mathbb{E}\left[\text{SUBOPT}_{\mathcal{S}_i}(I)\right] \leq c\mathcal{H}(t_i^D) = c\mathcal{H}(n_i)$. Therefore, for this case, $I$ is $c$-risk-competitive to Algorithm ERM.

In the other case, when $n_i > \rho i/2$, we have: $t_i^D = \rho i/2 = \left(\frac{\rho}{2\widetilde{\lambda}_i}\right) n_i$. Further, $\mathbb{E}\left[\text{SUBOPT}_{\mathcal{S}_i}(I)\right] \leq$
$c\mathcal{H}(t_i^D) = c\mathcal{H}(\frac{\rho}{2\widetilde{\lambda}_i} n_i) = c\left(\frac{2\widetilde{\lambda}_i}{\rho}\right)^{\alpha}\mathcal{H}(n_i)$  $\square$

**Discussion:** $\widetilde{\lambda}_i = (n_i/i)$ is the average rate of arrivals in a time step. We expect the ratio $(\widetilde{\lambda}_i/\rho)$ to be a small constant. If this ratio is a large number, much greater than 1, the total number of arrivals over $i$ time steps far exceeds the number of gradient computations the algorithm can perform over $i$ time steps. This rate of arrivals is unsustainable, because most practical algorithms such as SGD and variants, including SVRG and SAGA, require more than one gradient computation for each training point. Hence, the above lemma implies that if $I$ is $O(1)$-risk-competitive to DYNASAGA($\rho$), then it is also $O(1)$-risk-competitive to ERM, under reasonable arrival patterns.

Finally, we will bound the expected sub-optimality of STRSAGA over its effective sample set $T_i$ in Lemma 3. The proof of this lemma is presented in the supplementary material. In Section 5, we will show how to apply the following result to establish the risk-competitiveness of STRSAGA.

**Lemma 3.** *Suppose all $f_{\mathbf{x}}$ are convex and their gradients are L-Lipschitz continuous, and that $\mathcal{R}_{T_i}$ is $\mu$-strongly convex. At the end of each time step $i$, the expected sub-optimality of STRSAGA over $T_i$ is*

$$\mathbb{E}\left[\text{SUBOPT}_{T_i}(\text{STRSAGA})\right] \leq \mathcal{H}(t_i^{\text{STR}}) + 2\left(\mathcal{R}(\mathbf{w}_0) - \mathcal{R}(\mathbf{w}^*)\right)\left(\frac{L}{\mu}\right)^3\left(\frac{1}{t_i^{\text{STR}}}\right)^2.$$

*If we additionally assume that the condition number $L/\mu$ is bounded by a constant at each time, the above simplifies to $\mathbb{E}\left[\text{SUBOPT}_{T_i}(\text{STRSAGA})\right] \leq (1 + o(1))\mathcal{H}(t_i^{\text{STR}})$.*

## 5 Competitive analysis of STRSAGA on specific arrival distributions

Lemma 3 shows that the expected sub-optimality of STRSAGA over its effective sample set $T_i$ is $O(\mathcal{H}(t_i^{\text{STR}}))$ (note $t_i^{\text{STR}}$ is not equal to $n_i$ the number of points so far). However, our goal is to show that STRSAGA is risk-competitive to DYNASAGA($\rho$) at each time step $i$; i.e., the expected sub-optimality of STRSAGA over $\mathcal{S}_i$ is within a factor of $\mathcal{H}(t_i^D)$. The connection between the two depends on the relation between $t_i^{\text{STR}}$ and $t_i^D$. This relation is captured using sample-competitiveness, which is introduced in this section. Although not every arrival distribution provides sample-competitiveness, we will show a number of different patterns of arrival distributions that do provide this property. To model different arrival patterns, we consider a general arrival model where the number of points arriving in time step $i$ is a random variable $x_i$ which is independently drawn from distribution $\mathcal{P}$ with a finite mean $\lambda$. We consider arrival distributions of varying degrees of generality, including Poisson

arrivals, skewed arrivals, general arrivals with a bounded maximum, and general arrivals with an unbounded maximum. The proofs of results about specific distributions, as well as the full statements of prior theorems and bounds referenced below, can be found in the supplementary material.

**Definition 3.** *At time $i$, STRSAGA is said to be $k$-sample-competitive to DYNASAGA $(\rho)$ if $t_i^{\text{STR}}/t_i^D \geq k$.*

**Lemma 4.** *If STRSAGA is $k$-sample-competitive to DYNASAGA $(\rho)$ at time step $i$, then it is $c$-risk-competitive to DYNASAGA $(\rho)$ at time step $i$ with $c = k^{-\alpha}(2 + o(1))$.*

*Proof.* Let $T_i^{\text{STR}}$ and $T_i^D$ denote the effective samples that were used at iteration $i$ for STRSAGA and DYNASAGA $(\rho)$, respectively. We know that $T_i^{\text{STR}}, T_i^D \subseteq S_i$. Using Theorem 3 from [DLH16], we have: $\mathbb{E}\left[\text{SUBOPT}_{S_i}(\text{STRSAGA})\right] \leq \mathbb{E}\left[\text{SUBOPT}_{T_i}(\text{STRSAGA})\right] + \frac{n_i - t_i^{\text{STR}}}{n_i}\mathcal{H}(t_i^{\text{STR}})$.

Using Lemma 3, we can rewrite the above inequality as:
$\mathbb{E}\left[\text{SUBOPT}_{S_i}(\text{STRSAGA})\right] \leq (1 + o(1))\mathcal{H}(t_i^{\text{STR}}) + \frac{n_i - t_i^{\text{STR}}}{n_i}\mathcal{H}(t_i^{\text{STR}}) \leq (2 + o(1))\mathcal{H}(t_i^{\text{STR}})$.

If STRSAGA is $k$-sample-competitive to DYNASAGA $(\rho)$, then we have: $\mathbb{E}\left[\text{SUBOPT}_{S_i}(\text{STRSAGA})\right] \leq (2 + o(1))\mathcal{H}(t_i^{\text{STR}}) \leq (2 + o(1))\mathcal{H}(k \cdot t_i^D) = k^{-\alpha}(2 + o(1))\mathcal{H}(t_i^D)$, completing the proof. $\square$

**Lemma 5.** *At time step $i$, suppose the streaming arrivals satisfy: $n_{i/2} \geq kn_i$. Then, STRSAGA is $\min\{k, \frac{1}{2}\}$-sample-competitive to DYNASAGA $(\rho)$ at time step $i$.*

*Proof.* We first bound $t_i^{\text{STR}}$. At time $i/2$, at least $kn_i$ points have arrived. In Algorithm 1, at time $i/2$, these points are either in the Buf or already in the effective sample $T_{i/2}^{\text{STR}}$. We note that for every two iterations of SAGA, the algorithm moves one point from Buf (if available) to the effective sample, thus increasing the size of the effective sample set by 1. In the $i/2$ time steps from $i/2 + 1, \ldots, i$, STRSAGA can perform $\rho i/2$ iterations of SAGA. Within these iterations, it can move $\rho i/4$ points to $T_i^{\text{STR}}$, if available in the buffer. Hence, the effective sample size for STRSAGA at time $i$ is: $t_i^{\text{STR}} \geq \min\{\rho i/4, kn_i\}$. We know $t_i^D = \min\{n_i, \rho i/2\}$.

We consider four cases. In the first case, (1) if $\rho i/4 < n_{i/2}$ and $n_i < \rho i/2$, then $t_i^D = n_i$ and $t_i^{\text{STR}} \geq \rho i/4$. In this case, we have $t_i^{\text{STR}} \geq \rho i/4 > n_i/2 = t_i^D/2$. The other three cases, (2) $\rho i/4 < n_{i/2}$ and $n_i \geq \rho i/2$, (3) $\rho i/4 \geq n_{i/2}$ and $n_i < \rho i/2$, and (4) $\rho i/4 \geq n_{i/2}$ and $n_i \geq \rho i/2$, can be handled similarly. $\square$

**Skewed Arrivals with a Bounded Maximum.** We next consider an arrival distribution parameterized by integer $M \geq \lambda$, where the number of arrivals per time step can either be high ($M$) or zero. More precisely, $x_i = M$ with prob. $\frac{\lambda}{M}$ and $x_i = 0$ with prob. $1 - \frac{\lambda}{M}$. Thus, $E[x_i] = \lambda$. For $M > \lambda$, this models bursty arrival distributions with a number of "quiet" time steps with no arrivals, combined with an occasional burst of $M$ arrivals. We have the following result for skewed arrivals.

**Lemma 6.** *For a skewed arrival distribution with maximum $M$ and mean $\lambda$, STRSAGA is $6^\alpha(2 + o(1))$-risk-competitive to DYNASAGA $(\rho)$, with probability at least $1 - \epsilon$, at any time step $i > \frac{16M}{\lambda}\ln\frac{1}{\epsilon}$.*

At a high level, the proof relies on showing sample-competitiveness of STRSAGA. For a time step $i$ greater than the threshold stated in the lemma, we can prove the concentration of $n_i$ and $n_{i/2}$ using a Chernoff bound. Using Lemma 5, STRSAGA and DYNASAGA $(\rho)$ are $\frac{1}{6}$-sample-competitive, and the risk-competitiveness follows from Lemma 4. Note that as $M$ increases, arrivals become more bursty, and it takes longer for the algorithm to be competitive, with a high confidence.

**General Arrivals with a Bounded Maximum.** We next consider a more general arrival distribution with a maximum of $M$ arrivals, and a mean of $\lambda$. $x_i = j$ with probability $p_j$ for $j = 0, \ldots, M$, such that $\sum_j p_j = 1$ and $E[x_i] = \lambda$, for an integer $M > 0$.

**Lemma 7.** *For a general arrival distribution with mean $\lambda$ and maximum $M$, at any time step $i > \left(\frac{16M}{\lambda} + \frac{8}{3}\right)\ln\frac{1}{\epsilon}$, STRSAGA is $8^\alpha(2 + o(1))$-risk-competitive to DYNASAGA $(\rho)$, with probability at least $1 - \epsilon$.*

The high-level proof sketch for this case is similar to the case of skewed arrivals. The technical aspect is that in order to prove concentration bounds for $n_i$ and $n_{i/2}$, we use Bernstein's inequality [Mas07], which lets us bound the sum of independent random variables in a more flexible manner than Chernoff bounds (for random variables that are not necessarily binary valued), in conjunction with a bound on the variance of the distribution. Proof details are in the supplementary material.

**General Arrivals with an Unbounded Maximum.** More generally, the number of arrivals in a time step may not have a specified maximum. The arrival distribution can have a finite mean, despite a

small probability of reaching arbitrarily large values. We consider a sub-class of such distributions where all the polynomial moments are bounded, as in the following *Bernstein's condition* with parameter $b$: The random variable $x_i$ has mean $\lambda$, variance $\sigma^2$, and $\left|\mathbb{E}\left[(x_i - \lambda)^k\right]\right| \leq \frac{1}{2}k!\sigma^2 b^{k-2}$ for all integers $k \geq 3$ [Mas07].

**Lemma 8.** *For any arrival distribution with mean $\lambda$, bounded variance $\sigma^2$ and satisfying Bernstein's condition with parameter $b$, STRSAGA is $8^\alpha(2 + o(1))$-risk-competitive to DYNASAGA ($\rho$), with probability at least $1 - \epsilon$, at any time step $i > \max((16(\frac{\sigma}{\lambda})^2 + \frac{8}{3})\ln\frac{1}{\epsilon}, 2((\frac{\sigma}{\lambda})^2 + \frac{b}{\lambda})\ln\frac{1}{\epsilon})$.*

**Poisson Arrivals.** We next consider the case where the number of points arriving in each time step follows a Poisson distribution with mean $\lambda$, i.e., $\Pr[x_i = k] = \frac{e^{-\lambda}\lambda^k}{k!}$ for integer $k \geq 0$.

**Lemma 9.** *For Poisson arrival distribution with mean $\lambda$, STRSAGA is $8^\alpha(2 + o(1))$-risk-competitive to DYNASAGA ($\rho$) with probability at least $1 - \epsilon$, at any time step $i > \frac{16}{\lambda}\ln\frac{1}{\epsilon}$.*

The proof depends on a version of the Chernoff bounds tailored to the Poisson distribution—further details are in the supplementary material.

# 6 Experimental results

We empirically confirm the competitiveness of STRSAGA with the offline algorithm DYNASAGA($\rho$) through a set of experiments on real world datasets streamed in under various arrival distributions. We consider two optimization problems that arise in supervised learning, logistic regression (convex) and matrix factorization (nonconvex). For logistic regression, we use the A9A [DKT17] and RCV1.binary [LYRL04] datasets, and for matrix factorization, we use two datasets of user-item ratings from Movielens [HK16]. More detail on the datasets are provided in the supplementary material. These static training data are converted into streams, by ordering them by a random permutation, and defining an arrival rate $\lambda$ dependent on the dataset size. In our experiments, the training data arrives over the course of 100 time steps, with skewed arrivals parameterized by $M = 8\lambda$. Experiments on Poisson arrivals are given in the supplementary material.

At each time step $i$, a streaming data algorithm has access to $\rho$ gradient computations to update the model; we show results for $\rho/\lambda = 1$ and $\rho/\lambda = 5$. We compare the sub-optimality of STRSAGA with the offline algorithm DYNASAGA($\rho$), which is run from scratch at each time $i$ using $\rho i$ steps on $S_i$. We also compare with two streaming data algorithms, SGD, and for the case $\rho/\lambda = 1$, the single-pass algorithm SSVRG.[4] In the streaming data setting, in which we are not limited in storage and the available processing time $\rho$ may permit revisiting points, our implementation of SGD needs clarification in its sampling procedure. We tried two sampling policies. In the first, at each time step $i$ we sample points uniformly from $S_i$, the set of all points received till time step $i$. In the second, at each time step $i$ we first visit points in $S_i$ that have not been seen yet, and spend any remaining processing time to sample uniformly from all of $S_i$. In every case, the second method was better or indistinguishable from the first, and so all of our results are based on the second method. For our implementation of SSVRG, we have relaxed the memory limitation of the original streaming algorithm by introducing a buffer to store points that have arrived but not yet been processed. With this additional storage, we allow SSVRG to make progress during time steps even when no new points arrive, and hence make for a fairer comparison when data points do not arrive at a steady rate.

The main results are summarized in Figure 2, showing the sub-optimality of each algorithm and the sample-competitive ratio for STRSAGA. Additional plots of the test loss are given in the supplementary material. The dips in the sample-competitive ratio represent the arrival of a large group of points, and correspondingly at those times, the sub-optimality spikes, because there are now many new points added to $S_i$ that have yet to be processed. We observe that the sample-competitive ratio improves over the lifetime of the stream and tends towards 1, outperforming our pessimistic theoretical analysis. Furthermore, as the sample-competitive ratio increases, the risk-competitiveness of STRSAGA improves so that the sub-optimality of STRSAGA is comparable to that of the offline DYNASAGA($\rho$), which is the best we can do given limited computational power. In Figure 2, we also observe that STRSAGA outperforms both our streaming data version of SGD, due to the faster convergence rate when using SAGA steps with reduced variance, and also SSVRG, showing the benefit of revisiting data points, even when the processing rate is constrained at $\rho = 1\lambda$.

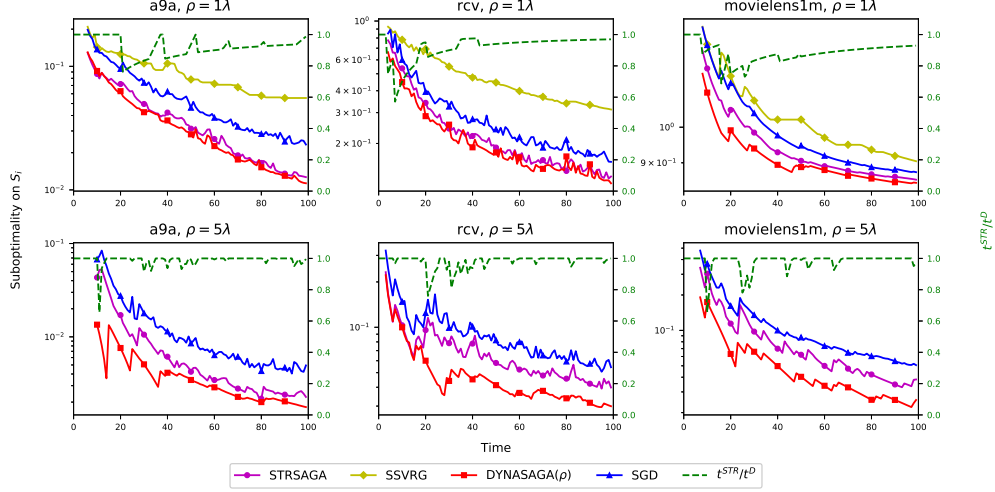

Figure 2: Sub-optimality under skewed arrivals with $M = 8\lambda$. Top row is processing rate $\rho = 1\lambda$, and bottom row is $\rho = 5\lambda$. The median is taken over 5 runs.

To better understand the impact of the skewed arrival distribution on the performance of STRSAGA, we did three experiments in Figure 3, showing the following results. (1) As $M/\lambda$ increases, the arrivals become more bursty and it takes longer for STRSAGA to be sample-competitive, and as a result, risk-competitive to DYNASAGA($\rho$). Note that the far left endpoint, for skewed arrival parameterized with $M = \lambda$, is the case of constant arrivals. (2) We observe that there is an intermediate point for $\rho/\lambda$ where it is more difficult to be sample-competitive, but at the extremes the ratio tends towards 1. This is because for large $\rho/\lambda$, whenever a big group of points arrives they can all be processed quickly. On the other hand, for small $\rho/\lambda$, at any time $i$, both STRSAGA and the offline algorithm are still processing points that arrived at some time significantly before $i$, and so a large variance in the amount of fresh arrivals at the tail of the stream can be tolerated. (3) The bound on sub-optimality we showed earlier is dependent on the number of data points processed so far. As we see, as time passes and STRSAGA sees more data points, its sub-optimality on $\mathcal{S}_i$ improves. Additionally as $\rho/\lambda$ increases, STRSAGA has more steps available to incorporate newly arrived data points and becomes more resilient to bursty arrivals.

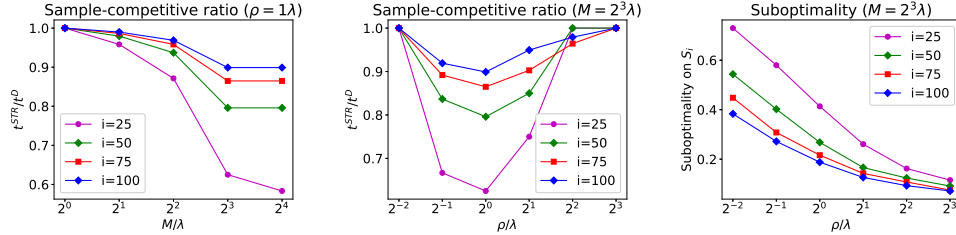

Figure 3: Sensitivity analysis. The first plot varies the skew $M/\lambda$ for a fixed processing rate $\rho/\lambda$, and the second two plots vary the processing rate for a fixed skew. Results are plotted for time steps $i = 25, 50, 75, 100$ over a stream of the RCV dataset of 100 time steps. The median is taken over 9 runs.

## 7 Conclusion

We considered the ongoing maintenance of a model over data points that are arriving over time, according to an unknown arrival distribution. We presented STRSAGA, and showed through both analysis and experiments that, for various arrival distributions, (i) its empirical risk over the data arriving so far is close to the empirical risk minimizer over the same data, (ii) it is competitive with a state-of-the-art offline algorithm DYNASAGA, and (iii) it significantly outperforms streaming data versions of both SGD and SSVRG. We conclude that STRSAGA should be the algorithm of choice for variance-reduced SGD on streaming data in the setting where memory is not limited.

## Footnotes

[4]We consider SSVRG a $\rho/\lambda = 1$ algorithm, because for most data points it receives, it uses 1 gradient computation, and only for an $o(1)$ fraction of the data points does it require 2 gradient computations.

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
