[Supplementary Material]

# A  Proofs from the analysis of STRSAGA

This section contains proof details from the analysis of STRSAGA. Throughout, we will assume that all $f_{\mathbf{x}}$ are convex and their gradients are $L$-Lipschitz continuous, and that $\mathcal{R}_{\mathcal{S}}$ is $\mu$-strongly convex for the set of training samples $\mathcal{S}$. In addition, we will use $\mathbf{U}$ which we define as follow:

$$\mathbf{U}(t, n) = \min \begin{cases} \rho_n \mathbf{U}(t - 1, n) \\ \min_{m < n}[\mathbf{U}(t, m) + \dfrac{n - m}{n} \mathcal{H}(m)], \end{cases} \quad (1)$$

where $\rho_n$ is the convergence rate of SAGA and defines as $\rho_n = 1 - \min(\frac{1}{n}, \frac{\mu}{L})$. And, the initial error $\mathbf{U}(0, m) = \zeta$ is defined as:

$$\zeta := \frac{4L}{\mu}[\mathcal{R}(\mathbf{w}_0) - \mathcal{R}(\mathbf{w}^*)].$$

We will use the following results from [DLH16].

**Lemma 10.** (THEOREM 3 IN [DLH16]) *Suppose the expected sub-optimality of an algorithm $A$ over a training set $\mathcal{T} \subseteq \mathcal{S}$ is bounded as $\mathbb{E}\left[\text{SUBOPT}_{\mathcal{T}}(A)\right] \leq \epsilon$. Then the expected sub-optimality of $A$ over $\mathcal{S}$ is bounded by $\mathbb{E}\left[\text{SUBOPT}_{\mathcal{S}}(A)\right] \leq \epsilon + \frac{n-m}{n}\mathcal{H}(m)$, where $|\mathcal{T}| = m, |\mathcal{S}| = n$.*

**Lemma 11.** (PROPOSITION 4. IN [DLH16]) *The expected sub-optimality of DYNASAGA over a training set $\mathcal{S}$ at iteration $t$ is*

$$\mathbb{E}\left[\text{SUBOPT}_{\mathcal{S}}(\text{DYNASAGA})\right] \leq \mathbf{U}(t, n).$$

*where the expectation is taken over randomness of $\mathcal{S}$, and $\mathbf{U}$ is defined in (1).*

**Lemma 12.** (LEMMA 5 IN [DLH16]) *For $\mathcal{H}(n) = cn^{-\alpha}$, where $1/2 \leq \alpha \leq 1$,*

$$\mathbf{U}(2n, n) \leq \mathcal{H}(n) + \frac{\zeta}{2}\left(\frac{L}{\mu n}\right)^2.$$

Using above results, we will bound the expected sub-optimality of STRSAGA in terms of the $\mathbf{U}$ function.

**Lemma 13.** *At the end of each time step $i$, the expected sub-optimality of STRSAGA over $T_i$ is*

$$\mathbb{E}\left[\text{SUBOPT}_{T_i}(\text{STRSAGA})\right] \leq \mathbf{U}(2t_i^{\text{STR}}, t_i^{\text{STR}}).$$

*where $\mathbf{U}$ is the upper bound function defined in (1).*

*Proof.* The proof is similar to the proof of Proposition 4 in [DLH16]. Note that performing extra steps of SAGA when the Buf is empty does not weaken the bound. □

Next, we prove the main result we stated in Section 4 bounding the expected sub-optimality of STRSAGA by the $\mathcal{H}$ function of its effective sample set size.

**Lemma 3.** *Suppose all $f_{\mathbf{x}}$ are convex and their gradients are $L$-Lipschitz continuous, and that $\mathcal{R}_{T_i}$ is $\mu$-strongly convex. At the end of each time step $i$, the expected sub-optimality of STRSAGA over $T_i$ is*

$$\mathbb{E}\left[\text{SUBOPT}_{T_i}(\text{STRSAGA})\right] \leq \mathcal{H}(t_i^{\text{STR}}) + 2\left(\mathcal{R}(\mathbf{w}_0) - \mathcal{R}(\mathbf{w}^*)\right)\left(\frac{L}{\mu}\right)^3\left(\frac{1}{t_i^{\text{STR}}}\right)^2.$$

*If we additionally assume that the condition number $L/\mu$ is bounded by a constant at each time, the above simplifies to $\mathbb{E}\left[\text{SUBOPT}_{T_i}(\text{STRSAGA})\right] \leq (1 + o(1))\mathcal{H}(t_i^{\text{STR}})$.*

*Proof.* The expected sub-optimality is bounded by the $\mathbf{U}$ function by Lemma 13, and we have a bound on $\mathbf{U}$ by Lemma 12. Therefore,

$$\mathbb{E}\left[\text{SUBOPT}_{T_i}(\text{STRSAGA})\right] \leq \mathbf{U}(2t_i^{\text{STR}}, t_i^{\text{STR}})$$

$$\leq \mathcal{H}(t_i^{\text{STR}}) + \frac{\zeta}{2}\left(\frac{L}{\mu t_i^{\text{STR}}}\right)^2$$

$$= \mathcal{H}(t_i^{\text{STR}}) + 2\left(\mathcal{R}(\mathbf{w}_0) - \mathcal{R}(\mathbf{w}^*)\right)\left(\frac{L}{\mu}\right)^3\left(\frac{1}{t_i^{\text{STR}}}\right)^2.$$

□

In addition, we have the following result on the expected sub-optimality of `DYNASAGA`$(\rho)$.

**Lemma 14.** *At the end of each time step $i$, the expected sub-optimality of `DYNASAGA`$(\rho)$ over $\mathcal{S}_i$ is*

$$\mathbb{E}\left[\texttt{SUBOPT}_{\mathcal{S}_i}(\texttt{DYNASAGA}(\rho))\right] \leq \left(\max\left(1,\left(\frac{2\widetilde{\lambda}_i}{\rho}\right)^{1+\alpha}\right) + o(1)\right)\mathcal{H}(n_i)$$

*where $\widetilde{\lambda}_i = \left(\frac{n_i}{i}\right)$ and $n_i = |\mathcal{S}_i|$.*

*Proof.* According to Lemma 11, the expected sub-optimality of `DYNASAGA`$(\rho)$ over the sample set $\mathcal{S}_i$ of size $n_i$ after $t$ iterations is bounded by $\mathbf{U}(t,n_i)$. As mentioned earlier, the Algorithm `DYNASAGA`$(\rho)$ has limited computational power and can performs only $\rho i$ steps of SAGA. Thus,

$$\mathbb{E}\left[\texttt{SUBOPT}_{\mathcal{S}_i}(\texttt{DYNASAGA}(\rho))\right] \leq \mathbf{U}(\rho i, n_i)$$

If $\widetilde{\lambda}_i \leq \rho/2$, then

$$\mathbb{E}\left[\texttt{SUBOPT}_{\mathcal{S}_i}(\texttt{DYNASAGA}(\rho))\right] \leq \mathbf{U}(\rho i, n_i)$$
$$\leq \mathbf{U}(2\widetilde{\lambda}_i i, n_i) = \mathbf{U}(2n_i, n_i)$$
$$\leq \mathcal{H}(n_i) + \frac{\zeta}{2}\left(\frac{L}{\mu n_i}\right)^2$$

If $\widetilde{\lambda}_i > \rho/2$, then $n_i = (\widetilde{\lambda}_i)i > (\rho/2)\,i$. Let $T$ be a subset of $\mathcal{S}_i$ such that $|T| = (\rho/2)\,i$, then Lemma 10 results

$$\mathbb{E}\left[\texttt{SUBOPT}_{\mathcal{S}_i}(\texttt{DYNASAGA}(\rho))\right] \leq \mathbb{E}\left[\texttt{SUBOPT}_{T_i}(\texttt{DYNASAGA}(\rho))\right] + \frac{\widetilde{\lambda}_i i - (\rho/2)\,i}{(\rho/2)\,i}\mathcal{H}((\rho/2)i)$$
$$\leq \mathbf{U}(\rho i, (\rho/2)\,i) + \frac{\widetilde{\lambda}_i i - (\rho/2)\,i}{(\rho/2)\,i}\mathcal{H}((\rho/2)\,i)$$
$$\leq \mathcal{H}((\rho/2)\,i) + \frac{\zeta}{2}\left(\frac{L}{\mu(\rho/2)i}\right)^2 + \left(\frac{2\widetilde{\lambda}_i}{\rho} - 1\right)\mathcal{H}((\rho/2)\,i)$$
$$= \left(\frac{2\widetilde{\lambda}_i}{\rho}\right)\mathcal{H}\left(\frac{\rho}{2\widetilde{\lambda}_i}n_i\right) + \frac{\zeta}{2}\left(\frac{L}{\mu(\rho/2)i}\right)^2$$
$$= \left(\frac{2\widetilde{\lambda}_i}{\rho}\right)^{1+\alpha}\mathcal{H}(n_i) + \frac{\zeta}{2}\left(\frac{2\widetilde{\lambda}_i}{\rho}\right)^2\left(\frac{L}{\mu n_i}\right)^2$$

$\square$

# B  Proofs from competitive analysis of `STRSAGA` on specific arrival distributions

This section contains proof details from the competitive analysis of `STRSAGA` on specific arrival distributions. Throughout, we will assume that all $f_{\mathbf{x}}$ are convex and their gradients are $L$-Lipschitz continuous, and that $\mathcal{R}_{\mathcal{S}}$ is $\mu$-strongly convex for the set of training samples $\mathcal{S}$. In addition, we assume that the condition number $L/\mu$ is bounded by a constant at each time.

**Constant Arrival Rate.** We first consider the case where $x_i = \lambda$ for each $i$, so that the number of arrivals in each time step is the same.

**Lemma 15.** *For a constant arrival rate, `STRSAGA` is $(1 + o(1))$-risk-competitive to `DYNASAGA`$(\rho)$ at any time step.*

*Proof.* If $\rho/2 \leq \lambda$, for each time $i$, we have $t_i^{\texttt{STR}} = t_i^D = \rho i/2$. Similarly, if $\lambda < \rho/2$, we have $t_i^{\texttt{STR}} = t_i^D = \lambda i$. Using Lemma 3, we have: $\mathbb{E}\left[\texttt{SUBOPT}_{T_i}(\texttt{STRSAGA})\right] \leq (1 + o(1))\mathcal{H}(t_i^{\texttt{STR}}) = (1 + o(1))\mathcal{H}(t_i^D)$. Note that $(1 + o(1))\mathcal{H}(t_i^D)$ is identical to the upper bound that we get for the expected sub-optimality of `DYNASAGA`$(\rho)$. $\square$

We use the following Theorem 1 from [MU17].

**Theorem 1.** *(Chernoff Bound) Let $X_1, ..., X_n$ be independent Poisson trials such that $\Pr[X_i] = p_i$.*
*Let $X = \sum_{i=1}^{n} X_i$ and $\mu = \mathbb{E}[X]$. Then the following Chernoff bounds hold:*

- *For $0 < \delta < 1$,*
$$\Pr[X \le (1 - \delta)\mu] \le e^{-\mu\delta^2/2}$$

- *For $0 < \delta \le 1$,*
$$\Pr[X \ge (1 + \delta)\mu] \le e^{-\mu\delta^2/3}$$

**Lemma 16.** *For a skewed arrival distribution with mean $\lambda$ and parameterized by $M$, for $i > \frac{3M}{\delta^2\lambda}\ln\frac{1}{\epsilon}$,*
*with probability at least $1 - \epsilon$, we have $n_i \le (1 + \delta)\lambda i$, where $0 < \delta \le 1$.*

*Proof.* Let $Y_i$ denotes the number of non-empty arrivals in time steps $1, \ldots, i$. $Y_i$ follows the binomial distribution with parameters $n = i$, and $p = \lambda/M$, i.e., $Y_i \sim B(i, \lambda/M)$ and $\mathbb{E}[Y_i] = \lambda i/M$. According to Theorem 1, for $0 < \delta \le 1$:

$$\Pr[Y_i \ge (1 + \delta)\lambda i/M] \le e^{-\frac{\delta^2 \lambda i}{3M}} \le \epsilon$$

On the other hand, we have $n_i$, the number of arrivals in the first $i$ time steps, is $M \cdot Y_i$. Thus, for $i > \frac{3M}{\delta^2\lambda}\ln\frac{1}{\epsilon}$ with probability at least $1 - \epsilon$, we have $n_i \le (1 + \delta)\lambda i$. $\square$

**Lemma 17.** *For a skewed arrival distribution with mean $\lambda$ and parameterized by $M$, for $i > \frac{4M}{\delta^2\lambda}\ln\frac{1}{\epsilon}$,*
*with probability at least $1 - \epsilon$, we have $n_{i/2} \ge (1 - \delta)\lambda i/2$, where $0 < \delta < 1$.*

*Proof.* Same as Lemma 16, let $Y_i$ denotes the number of non-empty arrivals in time steps $1, \ldots, i$. $Y_i$ follows the binomial distribution with parameters $n = i$, and $p = \lambda/M$, i.e., $Y_i \sim B(i, \lambda/M)$ and $\mathbb{E}[Y_i] = \lambda i/M$. According to Theorem 1, for $0 < \delta < 1$:

$$\Pr\left[Y_{i/2} \le (1 - \delta)\frac{\lambda i}{2M}\right] \le e^{-\frac{\delta^2 \lambda i}{4M}} \le \epsilon$$

On the other hand, we have $n_{i/2}$, total number of arrivals in the first $i/2$ time steps, is $M \cdot Y_{i/2}$. Thus, for $i > \frac{4M}{\delta^2\lambda}\ln\frac{1}{\epsilon}$ with probability at least $1 - \epsilon$, we have $n_{i/2} \ge (1 - \delta)\lambda i/2$. $\square$

**Lemma 6.** *For a skewed arrival distribution with maximum $M$ and mean $\lambda$, STRSAGA is $6^\alpha(2+o(1))$-*
*risk-competitive to DYNASAGA ($\rho$), with probability at least $1 - \epsilon$, at any time step $i > \frac{16M}{\lambda}\ln\frac{1}{\epsilon}$.*

*Proof.* By setting $\delta = 1/2$ in Lemma 16, for $i > \frac{12M}{\lambda}\ln\frac{1}{\epsilon}$, with probability at least $1 - \epsilon$, we have $n_i \le \left(\frac{3}{2}\right)\lambda i$. On the other hand, by setting $\delta = 1/2$ in Lemma 17, for $i > \frac{16M}{\lambda}\ln\frac{1}{\epsilon}$, with probability at least $1 - \epsilon$, we have $n_{i/2} \ge \lambda i/4$. Therefore, using union bound we can conclude with probability at least $1 - 2\epsilon$, we have $n_{i/2} \ge \frac{1}{6}n_i$ for $i > \frac{16M}{\lambda}\ln\frac{1}{\epsilon}$. As a result, using Lemma 5, STRSAGA and DYNASAGA ($\rho$) are at least $\frac{1}{6}$-sample-competitive and therefore by Lemma 4, STRSAGA is $6^\alpha(2 + o(1))$-risk-competitive with DYNASAGA ($\rho$). $\square$

**Observation 1.** *Let $x_1, x_2, \ldots, x_n$ be independent random variables such that $\mathbb{E}[x_i] = \lambda$ and the*
*range of these random variables is $\{0, 1, , \ldots, M\}$, then the variance of $x_i$ is no more than $\lambda(M - \lambda)$.*

*Proof.*

$$Var[x_i] = \sum_{j=0}^{M} p_j(j - \lambda)^2 = \left(\sum_{j=0}^{M} j^2 p_j\right) - \lambda^2$$

$$\le M \sum_{j=0}^{M} j p_j - \lambda^2 = M\lambda - \lambda^2$$

$\square$

We use the following Theorem 2 from [Mas07].

**Theorem 2.** *(Bernstein's Inequality) Let $x_1, x_2, \ldots, x_n$ be independent bounded random variables such that $\mathbb{E}[x_i] = 0$ and $x_i \leq M$ with probability 1 and let $\sigma^2 = \frac{1}{n} \sum_{i=1}^{n} Var[x_i]$. Then for any $a \geq 0$ we have:*

$$\Pr\left[\frac{1}{n}\sum_{j=1}^{n} x_i \geq a\right] \leq e^{-\frac{na^2}{2\sigma^2 + 2Ma/3}}$$

Using Theorem 2, we can show:

**Lemma 18.** *For any general arrival distribution with mean $\lambda$ and bounded maximum $M$, for $i > \frac{2(k+2)}{3(k-1)^2} \frac{M}{\lambda} \ln \frac{1}{\epsilon}$, with probability at least $1 - \epsilon$, we have $n_i \leq k\lambda i$, for any $k > 1$.*

*Proof.* According to Observation 1, we have $Var[x_i] \leq M\lambda$. Let's define random variable $z_i = x_i - \lambda$. We have $\mathbb{E}[z_i] = 0$ and $Var[z_i] = Var[x_i] \leq M\lambda$. Now, using Theorem 2 (Bernstein's inequality) and setting $a$ to $\lambda$ we have:

$$\Pr[n_i \geq k\lambda i] = \Pr\left[\frac{1}{i}(n_i - \lambda i) \geq (k-1)\lambda\right] = \Pr\left[\frac{1}{i}\sum_{j=1}^{i}(x_j - \lambda) \geq (k-1)\lambda\right]$$

$$= \Pr\left[\frac{1}{i}\sum_{j=1}^{i} z_j \geq (k-1)\lambda\right] \leq e^{-\frac{i(k-1)^2\lambda^2}{2M\lambda + 2M(k-1)\lambda/3}} = e^{-\frac{3(k-1)^2}{2(k+2)}\frac{\lambda}{M}i}$$

For $i > \frac{2(k+2)}{3(k-1)^2} \frac{M}{\lambda} \ln \frac{1}{\epsilon}$, this probability is at most $\epsilon$. $\qquad \square$

We use the following Theorem 3 from [BDR15].

**Theorem 3.** *Let $x_1, x_2, \ldots, x_n$ be a finite sequence of independent and non-negative random variables with finite variances. Denote $S_n = x_1 + x_2 + \ldots + x_n$ and $V_n = Var(S_n)$. Then, for any $a \geq 0$,*

$$\Pr[S_n \leq \mathbb{E}[S_n] - a] \leq e^{-\frac{a^2}{2V_n + W_n}}$$

*where*

$$W_n = \frac{1}{3}\sum_{k=1}^{n}\left(\frac{m_k^2 - v_k}{m_k}\right)^2, \quad m_k = \mathbb{E}[x_k] \quad and \quad v_k = Var(x_k)$$

**Lemma 19.** *For any general arrival distribution with mean $\lambda$ and bounded maximum $M$, for $i > \frac{12M/\lambda + 2}{3(1-2k)^2} \ln \frac{1}{\epsilon}$, with probability at least $1 - \epsilon$, we have $n_{i/2} \geq k\lambda i$, for any $k < \frac{1}{2}$.*

*Proof.* According to Theorem 3:

$$\Pr\left[n_{i/2} \leq k\lambda i\right] = \Pr\left[n_{i/2} \leq \left(\lambda\frac{i}{2} - \lambda i(\frac{1}{2} - k)\right)\right] \leq e^{-\frac{a^2}{2V_n + W_n}} \leq e^{-\frac{\lambda^2 i^2(1/2 - k)^2}{\lambda Mi + \lambda^2 i/6}} = e^{-\frac{3(1-2k)^2}{12M/\lambda + 2}i}$$

Thus, for $i > \frac{12M/\lambda + 2}{3(1-2k)^2} \ln \frac{1}{\epsilon}$ with probability at least $1 - \epsilon$, we have $n_{i/2} \geq k\lambda i$. $\qquad \square$

**Lemma 7.** *For a general arrival distribution with mean $\lambda$ and maximum $M$, at any time step $i > \left(\frac{16M}{\lambda} + \frac{8}{3}\right)\ln \frac{1}{\epsilon}$, STRSAGA is $8^\alpha(2 + o(1))$-risk-competitive to DYNASAGA ($\rho$), with probability at least $1 - \epsilon$.*

*Proof.* Similar to the proof of Lemma 6, by setting $k = 2$ in Lemma 18 and $k = 1/4$ in Lemma 19.
$\qquad \square$

We use the following Theorem 4 from [Mas07].

**Theorem 4.** *Let $x_1, x_2, \ldots, x_n$ be independent random variables with mean $\lambda$ and variance $\sigma^2$ satisfying the Bernstein condition with parameter $b$, $\left| \mathbb{E}\left[ (x_i - \lambda)^k \right] \right| \le \frac{1}{2} k! \sigma^2 b^{k-2}$ for all integers $k \ge 3$. Then for any $t \ge 0$ we have:*

$$\Pr\left[ \sum_{i}^{n} (x_i - \lambda) \ge t \right] \le e^{-\frac{t^2}{2(\sigma^2 + bt)}}$$

Using above theorem we have:

**Lemma 20.** *For any arrival distribution $x_i$ with $\mathbb{E}\left[ x_i \right] = \lambda$ and variance $\sigma^2$ that satisfies Bernstein's condition with parameter $b$, for $i > \frac{2(\sigma^2 + b\lambda)}{(k-1)^2 \lambda^2} \ln \frac{1}{\epsilon}$, with probability at least $1 - \epsilon$, we have $n_i \le k\lambda i$, for any $k > 1$.*

*Proof.* According to Theorem 4, we have:

$$Pr[n_i \ge k\lambda i] = Pr[(n_i - \lambda i) \ge (k-1)\lambda i] \le e^{-\frac{(k-1)^2 \lambda^2}{2(\sigma^2 + b\lambda)} i}$$

when $i \ge \frac{2(\sigma^2 + b\lambda)}{(k-1)^2 \lambda^2} \ln \frac{1}{\epsilon}$, this probability is at most $\epsilon$. $\square$

**Lemma 21.** *For any arrival distribution $x_i$ with mean $\lambda$ and variance $\sigma^2$, for $i > \frac{12(\sigma/\lambda)^2 + 2}{3(1-2k)^2} \ln \frac{1}{\epsilon}$, with probability at least $1 - \epsilon$, we have $n_{i/2} \ge k\lambda i$ for any $k < \frac{1}{2}$.*

*Proof.* Similar to the proof of Lemma 19. $\square$

**Lemma 8.** *For any arrival distribution with mean $\lambda$, bounded variance $\sigma^2$ and satisfying Bernstein's condition with parameter $b$, STRSAGA is $8^\alpha(2 + o(1))$-risk-competitive to DYNASAGA $(\rho)$, with probability at least $1 - \epsilon$, at any time step $i > \max\left( (16(\frac{\sigma}{\lambda})^2 + \frac{8}{3}) \ln \frac{1}{\epsilon}, 2((\frac{\sigma}{\lambda})^2 + \frac{b}{\lambda}) \ln \frac{1}{\epsilon} \right)$.*

*Proof.* Similar to the proof of Lemma 6, by setting $k = 2$ in Lemma 20 and $k = 1/4$ in Lemma 21. $\square$

**Lemma 22.** *For $i > \frac{3}{\lambda \delta^2} \ln \frac{1}{\epsilon}$, with probability at least $1 - \epsilon$, we have $n_i \le (1 + \delta)\lambda i$ for any $0 < \delta \le 1$.*

*Proof.* It can be inferred from Theorem 1. $\square$

**Lemma 23.** *For $i > \frac{4}{\lambda \delta^2} \ln \frac{1}{\epsilon}$, with probability at least $1 - \epsilon$, we have $n_{i/2} \ge (1 - \delta)\lambda i / 2$ for any $0 < \delta < 1$.*

*Proof.* It can be inferred from Theorem 1. $\square$

**Lemma 9.** *For Poisson arrival distribution with mean $\lambda$, STRSAGA is $8^\alpha(2 + o(1))$-risk-competitive to DYNASAGA $(\rho)$ with probability at least $1 - \epsilon$, at any time step $i > \frac{16}{\lambda} \ln \frac{1}{\epsilon}$.*

*Proof.* Similar to the proof of Lemma 6, by setting $\delta = 1$ in Lemma 22 and $\delta = 1/2$ in Lemma 23. $\square$

## C  Additional experimental details

**Setup** All algorithms were implemented in Python using numpy, and the experiments were run on a 64-bit Intel(R) Xeon(R) CPU clocked at 3.30 GHz and 8G DDR3 RAM.

**Datasets** Details of the 4 real-world datasets we used are given in Tables 1 and 2. We reserve $10\%$ of each dataset for testing and use the remaining $90\%$ for training.

The loss function for the binary classification task is L2-regularized logistic loss. For a data point $(x, y)$, the corresponding loss is $f_{(x,y)}(\mathbf{w}) = \log(1 + \exp(-y\mathbf{w}^T x)) + \frac{\mu}{2} ||\mathbf{w}||_2^2$. For collaborative filtering, we solve the matrix factorization problem of finding two rank-10 matrices, $\mathbf{w} = (L, R)$, so that $LR^T$ approximates the known elements of the data matrix $M$. The regularized loss function for

Table 1: *Datasets for logistic regression*

| Dataset | Size | Number of Features |
|---|---|---|
| RCV1.BINARY | 20242 | 47236 |
| A9A | 32561 | 123 |

Table 2: *Datasets for matrix factorization*

| Dataset | Users | Movies | Date Range | Rating Scale | Density |
|---|---|---|---|---|---|
| MovieLens100K | 943 | 1682 | 9/1997-4/1998 | 1-5, stars | 6.30% |
| MovieLens1M | 6040 | 3706 | 4/2000-2/2003 | 1-5, stars | 4.47% |

the data point $M_{ij}$ is $f_{(i,j)}(\mathbf{w}) = ((LR^T)_{ij} - M_{ij})^2 + \frac{\mu}{2}(||L||_F^2 + ||R||_F^2)$. The rank 10 for matrix factorization was chosen for good validation set error after optimizing with SGD after a single pass over the static dataset. The setting of $\mu$ for each dataset similarly chosen to minimize the validation set error, $\mu_{\text{A9A}} = 10^{-3}, \mu_{\text{RCV}} = 10^{-5}, \mu_{\text{MovieLens}} = 10^{-1}$. The step sizes we used for each algorithm and each dataset were again chosen to minimize the validation set error after a single pass. For SGD, we used a constant step size, which performed better than a decaying step size of the form $\eta_t = \eta_0/(1 + \eta_0\mu t)$.

**Additional Results** In the main paper, we only showed the sub-optimality under skewed arrivals. In Figure 4, we plot the test loss. We observe that the accuracy of STRSAGA is comparable with the offline algorithm DYNASAGA($\rho$) under this bursty arrival pattern even at limited processing rates. Furthermore, STRSAGA yields a more accurate model than either SGD or SSVRG.

Figure 4: Test loss under skewed arrivals with $M = 8\lambda$. Top row is processing rate $\rho = 1\lambda$, and bottom row is $\rho = 5\lambda$. The median is taken over 5 runs.

We also consider Poisson arrivals. Sub-optimality is shown in Figure 5 and test loss in Figure 6. The median sample-competitive ratio is 1 from the beginning of the stream, which is significantly better than the ratio we showed analytically. Note that the curves for STRSAGA and DYNASAGA($\rho$) coincide for $\rho/\lambda = 1$ when STRSAGA is sample-competitive at all points in the stream, since the two algorithms are identical in this regime. Again we find that STRSAGA outperforms SGD and SSVRG.

All plots for the MovieLens100K dataset were omitted in the main paper. Sub-optimality is shown in Figure 7 and test loss in Figure 8. The trends are similar to those for the MovieLens1M dataset. One notable exception is the poorer performance of SSVRG. We have chosen similar hyperparameters for both the 100K and 1M datasets (in a streaming data setting, we generally do not know how much

Figure 5: Sub-optimality under Poisson arrivals with mean $\lambda$. Top row is processing rate $\rho = 1\lambda$, and bottom row is $\rho = 5\lambda$. The median is taken over 5 runs.

Figure 6: Test loss under Poisson arrivals with mean $\lambda$. Top row is processing rate $\rho = 1\lambda$, and bottom row is $\rho = 5\lambda$. The median is taken over 5 runs.

data will arrive in advance), and the slower convergence on the 100K dataset is likely due to a greater sensitivity to the hyperparameter selection of `SSVRG`.

Consistently we observe `STRSAGA` is increasingly close to the offline `DYNASAGA(`$\rho$`)` as time passes and that `STRSAGA` performs better than `SGD` and `SSVRG` across each dataset and both arrival distributions.

Figure 7: Additional plots of sub-optimality for MovieLens100k.

Figure 8: Additional plots of test loss for MovieLens100k.