[Reviews · NeurIPS 2018]

Reviewer 1



This paper considers the problem of streaming stochastic optimization taking into account the arrival patterns of examples in time. Whereas the relevant previous work focuses on learning from a stream (i.e. in one pass over the data, with O(dimension) memory), this work attempts to utilize the time spent between the arrival of examples in order to revisit previously encountered examples. The problem description is novel and appealing, both in its possible practical relevance and as an interesting new model for consideration in algorithm analysis. However, there are central issues with the comparisons that the paper draws between algorithms, both conceptually and experimentally. Conceptually, it seems incorrect to say that STRSAGA is a streaming algorithm, and in turn in to repeatedly compare it to one (such as SSVRG), since STRSAGA's memory complexity actually grows linearly with the dataset size (in order to maintain its "effective sample set"). In all previous related work, a streaming algorithm essentially means one whose memory usage does not grow with the dataset size. This is exactly what distinguishes streaming algorithms from ERM-based algorithms. The latter maintain an empirical sample of points in memory, whereas the former cannot maintain more than a constant number of them. Experimentally, the paper's main algorithm (STRSAGA) is missing some comparisons to natural baselines. Both DYNASAGA and SSVRG are not basic (nor widely used) algorithms. Meanwhile, a comparison to the simplest and most basic streaming algorithm, plain SGD, is missing. Relating to baselines, the paper includes the introductory remark (line 74): > We also show that STRSAGA significantly outperforms SSVRG [FGKS15], the state-of-the-art streaming data algorithm The original SSVRG paper [FGKS15] offers no experiments, and I don't know of any published since. It's a theoretical result, being used here as an experimental baseline, as a means of demonstrating an empirical advantage. Because it isn't actually a commonly accepted state of the art in the practical sense, outperforming SSVRG in practice does not imply an improvement relative to the state of scientific knowledge. Again, the more obvious comparison here would be to plain one-pass SGD. ---- Edited after author response: The authors addressed the concern about experimental comparisons to SGD in their response. I suggest including this in the paper. On the point of "streaming", here is a valid run of the algorithm: Fix an integer k. Suppose there is a dataset of size 2k, and STRSAGA is run for steps 1, ..., k, where at each step it receives two subsequent data points from the dataset. After step k, STRSAGA has half of the original dataset stored in its effective sample set. Now suppose STRSAGA continues to run for T additional steps, receiving zero new points at each step. Then, during these T additional steps, STRSAGA processes that half of the original dataset that it had stored. So STRSAGA's description includes, as a special case, an algorithm for minimizing empirical risk on half of a dataset, and this indicates that the paper needs work in its formal setup, not only in terminology. Maybe the right direction is to require a limit on memory, as reviewer 2 suggests as well.

Reviewer 2



In this paper, the concept of delayed arriving samples is combined with stochastic optimization. The interesting setup plotted here can be interpreted at controlling (or decision making or ) SGD considering the streaming arrival samples and computational limits. Authors proposed a streaming version of DYNASAGA (a stochastic optimization method with adaptive sample sizes) that adapts to their online setting. They introduce the notion of c-compatibility to compare their method to a computationally limited version of DYNASAGA (is called DYNASAGA(\rho)). Then dependency of compatibility factor to the average arrival time is an initiative result that is confirmed by their analysis. They also have proposed concrete examples of skewed and general arrivals that highlight the notion of compatibility (for sufficiently long time span) better. I’ve found experiments representative. In my opinion, the idea is novel and have potentials in online optimization. To improve the quality of the paper, however, I have the following suggestions: 1. In theoretical results of corollary 1, the dependency of the established bound to the condition L/mu should be expressed as the mu factor might be a function of the number of samples itself. 2. In corollary 1, I wondered whether the established upperbound should also depend on \rho. 3. My main concern about the algorithm is the memory issue. Since the setting is assumed to be online it might be reasonable to consider the memory limited and show how it can change the optimization plan. ------------------------ I have read the response.

Reviewer 3



The paper presents some interesting ideas and results. In general, I like the idea of the algorithm. STRSAGA uses the same idea as DYNASA. The different is that DYNASA is an offline algorithm while STRAGA also consider a streaming data with various arrival distribution. You have mentioned in Conclusion is that “its empirical risk over the data arriving so far is close to the empirical risk minimizer over the same data”. Do you have any explanation for this behavior? Do you think, because of this, they have a similar behavior as shown in experiments? (According to the experimental results, it seems that STRSAGA and DYNASA are not much different) For the theoretical part, since the results (for streaming data for various arrival distributions) are new, it would agree that it is not able to compare to SSVRG since SSVRG is not suited for these cases. I think this paper has some new ideas and may encourage people to investigate more in this direction. ------- AFTER REBUTTAL --------- The author(s) provided some experiments comparing to SSVRG that I requested. It seems a promising result to me. I think this paper has some new ideas and may encourage people to investigate more in this direction. I also like the idea of the algorithm. I vote for accepting this paper!